# High SARS-CoV-2 seroprevalence in Karaganda, Kazakhstan before the launch of COVID-19 vaccination

Irina Kadyrova[1]ᴼ*, Sergey Yegorov[2,3]ᴼ*, Baurzhan Negmetzhanov[3,4], Yevgeniya Kolesnikova[1], Svetlana Kolesnichenko[1], Ilya Korshukov[1], Lyudmila Akhmaltdinova[1], Dmitriy Vazenmiller[1], Yelena Stupina[1], Naylya Kabildina[1], Assem Ashimova[3], Aigul Raimbekova[3], Anar Turmukhambetova[1], Matthew S. Miller[2], Gonzalo Hortelano[3], Dmitriy Babenko[1]

1 Research Centre, Karaganda Medical University, Karaganda, Kazakhstan, 2 Michael G. DeGroote Institute for Infectious Disease Research, McMaster Immunology Research Centre, Department of Biochemistry and Biomedical Sciences, McMaster University, Hamilton, ON, Canada, 3 School of Sciences and Humanities, Nazarbayev University, Nur-Sultan, Kazakhstan, 4 National Laboratory Astana, Centre for Life Sciences, Nazarbayev University, Nur-Sultan, Kazakhstan

ᴼ These authors contributed equally to this work.
* ikadyrova@qmu.kz (IK); yegorovs@mcmaster.ca (SY)

**Data Availability Statement:** All relevant data are within the manuscript and its Supporting Information files.

## Abstract

COVID-19 exposure in Central Asia appears underestimated and SARS-CoV-2 seroprevalence data are urgently needed to inform ongoing vaccination efforts and other strategies to mitigate the regional pandemic. Here, in a pilot serologic study we assessed the prevalence of SARS-CoV-2 antibody-mediated immunity in a multi-ethnic cohort of public university employees in Karaganda, Kazakhstan. Asymptomatic subjects (n = 100) were recruited prior to their first COVID-19 vaccination. Questionnaires were administered to capture a range of demographic and clinical characteristics. Nasopharyngeal swabs were collected for SARS-CoV-2 RT-qPCR testing. Serological assays were performed to detect spike (S)-reactive IgG and IgA and to assess virus neutralization. Pre-pandemic samples were used to validate the assay positivity thresholds. S-IgG and -IgA seropositivity rates among SARS-CoV-2 PCR-negative participants (n = 100) were 42% (95% CI [32.2–52.3]) and 59% (95% CI [48.8–69.0]), respectively, and 64% (95% CI [53.4–73.1]) of the cohort tested positive for at least one of the antibodies. S-IgG titres correlated with virus neutralization activity, detectable in 49% of the tested subset with prior COVID-19 history. Serologically confirmed history of COVID-19 was associated with Kazakh ethnicity, but not with other ethnic minorities present in the cohort, and self-reported history of respiratory illness since March 2020. Overall, SARS-CoV-2 exposure in this cohort was ~15-fold higher compared to the reported all-time national and regional COVID-19 prevalence, consistent with recent studies of excess infection and death in Kazakhstan. Continuous serological surveillance provides important insights into COVID-19 transmission dynamics and may be used to better inform the regional public health response.

**Funding:** 1- Ministry of Education and Science of the Republic of Kazakhstan #AP09259123 Irina Kadyrova 2- Faculty Development Competitive Research Grant (COVID) of Nazarbayev University #280720FD1902 Gonzalo Hortelano 3- M.G. DeGroote Postdoctoral Fellowship, McMaster University Sergey Yegorov The funders of the study had no role in study design, data collection, data analysis, data interpretation, or writing of the report. All authors had full access to all the data in the study and the lead authors (IK, SY, DB) had final responsibility for the decision to submit manuscript for publication.

**Competing interests:** The authors have declared that no competing interests exist.

# Introduction

COVID-19 remains a global public health concern and is especially pernicious in regions with limited public health infrastructure that suffer from inadequate epidemiologic surveillance and delayed implementation of pandemic countermeasures. In the Central Asian states, such as Kazakhstan, substantial underestimations (of ~14-fold) of COVID-19 incidence and associated mortality [1–3] have led to public distrust and slow uptake of public health measures, including vaccination [4]. To date, the extent of community exposure to severe acute respiratory syndrome coronavirus 2 (SARS-CoV-2) in Kazakhstan is incompletely understood. Thus, as of 7 August 2021 (when analysis presented in this work was completed [5]), the officially reported number of all-time COVID-19 cases in Kazakhstan was 689,402 (626,402 of which were PCR-confirmed, while the rest were diagnosed based on clinical disease manifestations). These figures represented a cumulative prevalence of ~3.7% [6]- a prevalence that appears low given the substantial excess of infections and mortality estimated for Kazakhstan consistent with COVID-19 [2, 3].

Disparities between reported cases and true infections occur due to a plethora of factors, including unreported asymptomatic and mild infections, limited access to timely clinical and laboratory confirmation of COVID-19 diagnosis, and false-negative laboratory test results [7]. Case underestimation varies broadly by country and is most pronounced in lower income regions. Similar to its neighbouring Central Asian and Eastern European states, Kazakhstan's healthcare and vital registration systems have struggled to keep a consistent tally of COVID-19 incidence and mortality. This was especially evident at the onset of the pandemic in Spring-Summer of 2020 when cases were counted solely if COVID-19 was identified as the main cause of hospitalization and/or death, resulting in a significant undercounting of undiagnosed pneumonia cases, which most likely were COVID-19-associated [1, 3, 8].

One way to estimate the proportion of the population with previous exposure to COVID-19 is by using serological surveillance [7], which has been under-utilized in Kazakhstan and other Central Asian states. Here, we wished to gain insight into the true SARS-CoV-2 exposure rates in the Karaganda district of Kazakhstan, a multi-ethnic region, inhabited predominantly by ethnic Kazakhs (~60%), and other ethnic groups with diverse Slavic and Eastern European and Central Asian backgrounds. Therefore, we assessed full-length SARS-CoV-2 Spike (S)-specific IgG and IgA titres in a public university-based cohort, representing a diverse array of people with different risks of exposure to COVID-19.

# Materials and methods

## Study setting and participant recruitment

This study was conducted in conjunction with screening for a clinical trial assessing immunogenicity of the Sputnik-V vaccine (ClinicalTrials.gov #NCT04871841) based in Karaganda, the capital of Karaganda region situated in Central Kazakhstan [9]. This cohort was chosen for the serologic studies owing to funding availability and perceived feasibility in the context of the readily available resources and active participant recruitment within the infrastructure of the larger clinical trial.

Since February 2020 (and to the date when analysis presented in this work was completed), the Karaganda region (population ~1.3M) had over 63,000 (~5% of regional population) reported COVID-19 infections, placing it behind several other locales including the capital, Nur-Sultan (population ~ 1.0M), which has had a reported all-time COVID-19 prevalence of >11% [10]. Participant screening occurred in April-May 2021 at a COVID-19 vaccination clinic for university employees at the Karaganda Medical University. The study

participants comprised of medical university administrative staff and instructors (61%), clinical laboratory staff (9%), and healthcare practitioners from affiliated teaching hospitals (30%). Consenting, asymptomatic adults, who had not previously received a COVID-19 vaccine, were invited to participate in the study. Exclusion criteria were presence of respiratory symptoms or laboratory-confirmed COVID-19 diagnosis within two weeks prior to the study. Short questionnaires addressing the participants' demographic background and recent history of COVID-19 exposure were administered. To validate the IgG and IgA assay positivity thresholds, we performed ELISA on pre-pandemic samples, consisting of archived plasma samples (n = 10, 3 men and 7 women, median age (IQR) = 48(34.3–55.5) collected in 2016 as part of clinical studies of colorectal cancer and pertaining to the cancer-free control group in the original study [11].

## Sample collection and processing

Nasopharyngeal swabs were collected following the national guidelines into DNA/RNA shield media (Zymo Research, Irvine, US). Blood (5 ml) was collected by venipuncture into EDTA tubes (Improvacuter, Gel & EDTA.K2, Improve Medical Instruments, Guangzhou, China) both in the pandemic and pre-pandemic studies. Blood plasma was isolated by centrifugation at $2,000 \times g$ for 10 minutes. All samples were stored at -80˚C prior to analyses.

## PCR screening for SARS-CoV-2

Total RNA was isolated from nasopharyngeal swabs by magnetic bead-based nucleic acid extraction (RealBest Sorbitus, Vector-Best, Novosibirsk, Russia) and used for SARS-CoV-2 real-time RT-PCR testing by the Real-Best RNA SARS-CoV-2 kit (Vector-Best, Novosibirsk, Russia) targeting the SARS-CoV-2 RdRp and N loci, following the manufacturer's protocol.

## IgG and IgA assays

SARS-CoV-2 S1 IgG and IgA ELISAs were performed using commercially available assays (Euroimmun Medizinische Labordiagnostika AG, Lübeck, Germany) on the Evolis 100 ELISA reader (Bio-Rad) according to the manufacturers' protocols. Optical density (OD) ratios were calculated as ratio of the OD reading for each sample to the reading of the kit calibrator at 450 nm. In the initial analysis, we used the Euroimmun-recommended OD ratio cutoff values for both IgG and IgA, which are "<0.8" for Ig-negative samples, "0.8–1.1" for Ig-borderline samples, and "> = 1.1" for Ig-positive samples. We noted that using the manufacturer's cutoff values: of all IgG "borderline" participants (n = 9), 7 (77.8%) were IgA+ (IgA OD ratio> = 1.1), 1 (11.1%) was IgA borderline and 1 (11.1%) was IgA negative, while of all IgA "borderline" participants (n = 6), 2 (30.0%) were IgG+ (IgG OD ratio> = 1.1), 1 (20.0%) was IgG borderline, and 3 (50.0%) were IgG negative.

The mean OD450 ratios of the pre-pandemic samples were 0.3 for both IgG and IgA. Therefore, we empirically assumed that ~99.7% of IgG- and IgA- negative samples would fall within 3 standard deviations of the mean, i.e. within OD450 ratios of 0.45 and 0.51 for IgG and IgA, respectively. Thus, for both IgG and IgA assays, we used the manufacturer-recommended threshold (0.8), which is conservatively above our calculated empiric negative thresholds, and considered all samples with OD ratios <0.8 and > = 0.8 as "negative" and "positive", respectively. Using this in-house threshold, we defined the "No Prior COVID" subjects as negative for both IgG and IgA (IgG-, IgA-) and the "Prior COVID" subjects as positive for either or both IgG and/or IgA (IgG+/-, IgA+/-).

### Surrogate SARS-CoV-2 neutralization assays

A virus neutralizing assay was performed using a commercially available kit (cPass SARS-CoV-2 Neutralization Antibody Detection Kit, #L00847-C, GenScript Biotech Co., Nanjing City, China) in a subset of 55 participants. The assay is designed to assess inhibition of the interaction between the recombinant SARS-CoV-2 receptor binding domain (RBD) fragment and the human ACE2 receptor protein (hACE2). Briefly, plasma samples and manufacturer-provided controls were pre-incubated with the horseradish peroxidase (HRP)-conjugated RBD at 37˚C for 30 min, and then added to the hACE-2 pre-coated plate for incubation at 37˚C for 15 min. After washing and incubation with the tetramethylbenzidine (TMB) substrate, the absorbance of the final solution was measured at 450 nm using the Evolis 100 ELISA reader (Bio-Rad). Quality control was done following the manufacturer's recommendations. Neutralization % was calculated by subtracting the negative control-normalized absorbance of the samples from 1 and multiplying it by 100%; a manufacturer-recommended cut-off of 30% was used for detectable SARS-CoV-2 neutralization activity.

### Statistical analysis

Demographic differences were assessed using the two-sided Mann-Whitney U test for age and BMI, Pearson $\chi2$ for sex, ethnicity and comorbidities and Fisher's exact test for self-reported history and workplace exposure. The 95% confidence intervals (CI) were calculated using the binomial "exact" method. Correlations among variables were explored using the Spearman rank test and lines of best fit were derived via linear regression.

### Ethics statement

All study procedures were approved by the Research Ethics Board of Karaganda Medical University under Protocol #45 from 06.04.2020. Written informed consent was obtained from all participants.

## Results

All 100 participants tested negative for SARS-CoV-2 by RT-qPCR at screening. Of 100 participants, 42 (42.0%; 95% CI [32.2–52.3]) were S-IgG+. Due to insufficient sample volume, we were unable to include two samples (one IgG+ and one IgG-) in IgA testing, thus out of the 98 participants tested for IgA, 58 (59.2%; 95% CI [48.8–69.0]) were S-IgA+ (Fig 1A). None of the pre-pandemic samples were positive for S-IgG or -IgA (Fig 1A).

When stratified by the presence/absence of both IgG and IgA, there were 37 (37.8%) IgG+/IgA+, 4 (4.1%) IgG+/IgA-, 21 (21.4%) IgG-/IgA+ and 36 (36.7%) IgG-/IgA- subjects (Fig 1B). Cumulatively there were 63.6% (63/99; 95% CI [53.4–73.1]) subjects positive for at least one of the antibodies (including one participant with a missing S-IgA test results); these subjects were defined as the "Prior COVID" group (Table 1).

To further characterize the serologic features of the cohort, we compared the levels of SARS-CoV-2 binding antibodies and the capacity of participant plasma to neutralize SARS-CoV-2 RBD-hACE2 interaction *in vitro*. We found that S-IgG and S-IgA levels correlated strongly across the cohort (r = 0.599, p<0.001, Fig 1B). The SARS-CoV-2 neutralization capacity was significantly higher in the Prior COVID group compared to the No Prior COVID group (p<0.001, Fig 1C). In the Prior COVID group, 17 out of 35 tested participants (48.6%) exhibited neutralization exceeding the assay positivity cut-off (Fig 1C). SARS-CoV-2 neutralization also significantly correlated with circulating S-reactive IgG in the Prior COVID group (Fig 1D).

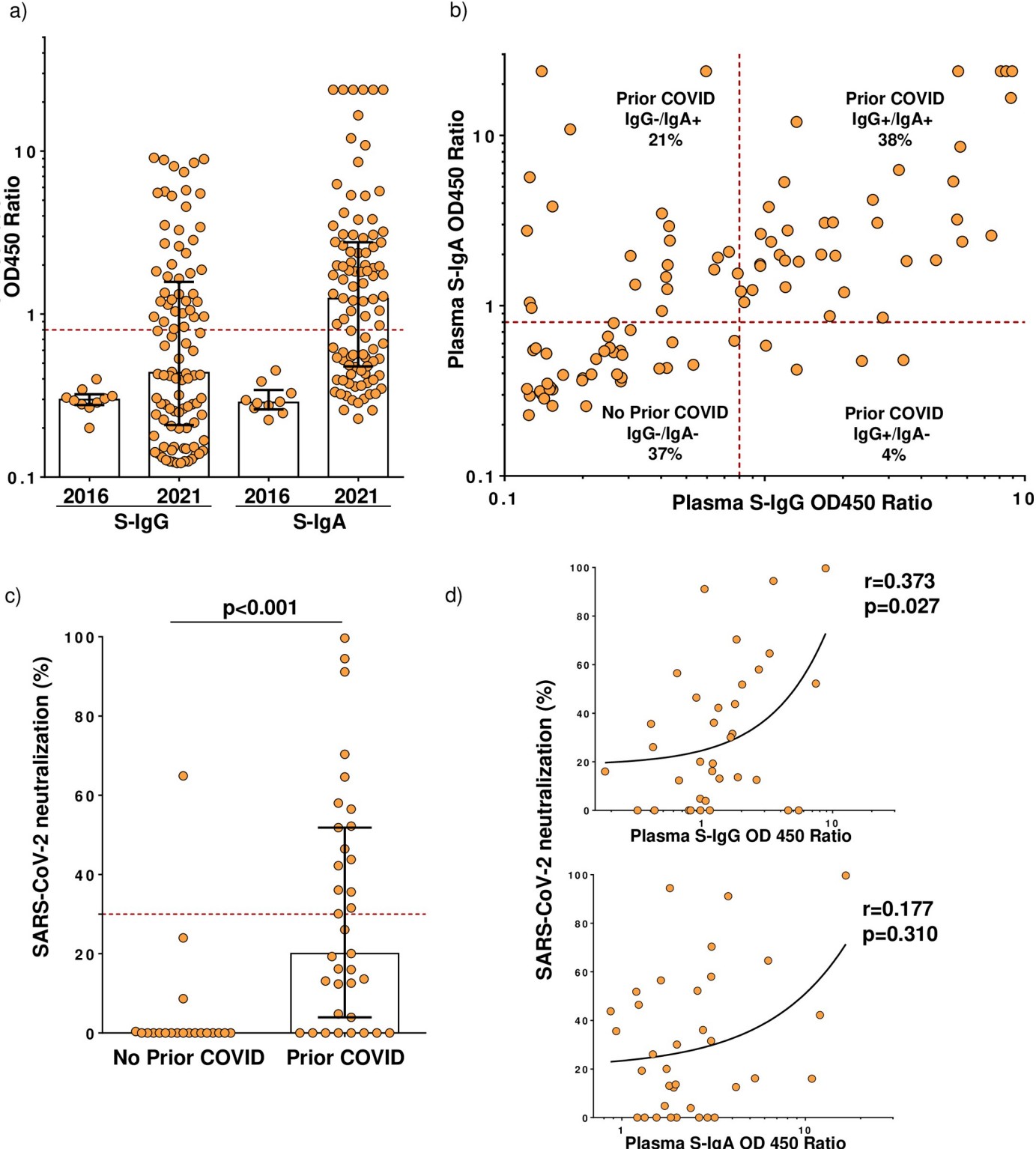

**Fig 1.** a) Distribution of optic density (OD) 450 ratios for blood SARS-CoV-2 Spike (S)-reactive IgG and IgA among the study participants recruited in Spring 2021 (n = 100) compared to the pre-pandemic samples obtained in 2016 (n = 10). b) Correlation plot of SARS-CoV-2 Spike-reactive IgG and IgA levels among the 2021 study participants (n = 98). In a) and b) the red dotted lines represent the assay cut-off values at OD450 ratios = 0.8 for positive samples for both IgG and IgA. c) The SARS-CoV-2 neutralization capacity of the 2021 study participant plasma samples measured using the surrogate virus neutralization test. N = 25 in the No Prior COVID group, and N = 30 in the Prior COVID group. The red dotted line represents the assay cut-off value at 30% for positive samples. d) Correlations

between SARS-CoV-2 neutralization versus S-reactive IgG and IgA among the study participants with a serology-confirmed history of COVID-19 (the Prior COVID-19 group, N = 30).

Lastly, we compared the clinical and demographic features of the cohort stratified by the serology-confirmed COVID-19 exposure. The Prior COVID group consisted of 25% more people self-identifying as ethnic Kazakhs (p = 0.022), and more frequently self-reported having had a respiratory illness since March 2020 (p<0.001), compared to the No prior COVID group (Table 1). Only in a minority of cases (24%, 8/33) self-reported respiratory illness was confirmed as COVID-19 by PCR and/or serology at the time of sickness; most of the self-reported respiratory illness occurred in March-Aug 2020, and in two cases in February-March 2021. There were no other significant clinical or demographic differences between the Prior COVID and No prior COVID groups (Table 1).

## Discussion

Here we present anti-S IgG and IgA-based seroprevalence findings from a cohort of public university employees in Karaganda, Kazakhstan, who were invited to participate in the study prior to receiving their first dose of COVID-19 vaccine. The cohort seropositivity for anti-S IgG and IgA was 41% and 59%, respectively, while 64% of the subjects tested positive for at least one of the antibodies, thus approaching the 67% seroprevalence threshold thought to be required for establishing herd immunity [12, 13]. Consistent with studies of excess infection and death in Kazakhstan [2, 3], the serologically assessed SARS-CoV-2 exposure in this cohort was 14-15-fold higher than the reported all-time national and regional COVID-19 prevalence.

The substantial discrepancy between serology-derived and officially reported COVID-19 exposure estimates is not uncommon across the globe [7]. However, what is most striking about our findings is the unusually high SARS-CoV-2 seroprevalence exceeding the estimates

**Table 1. Demographic and clinical characteristics of the study cohort.**

| | All (100) | No prior COVID (36*) | Prior COVID (63*) | P value |
|---|---|---|---|---|
| Age, years | 43.5 [35.3–54.0] | 43.0 [37.0–55.5] | 44.0 [34.3–55.5] | 0.657 |
| Sex (men) | 31.0% | 27.8% | 33.3% | 0.655 |
| Kazakh ethnicity~ | 55.0% | 38.9% | 63.5% | **0.022** |
| BMI#, kg/m² | 25.2 [22.8–28.1] | 24.8 [22.3–28.3] | 25.4 [23.4–28.2] | 0.440 |
| Any comorbidities^ | 45.0% | 41.7% | 46.0% | 0.834 |
| Self-reported history of respiratory illness since March 2020 | 37.0% | 9.1% | 60.0% | **<0.001** |
| *Potential workplace exposure to COVID-19* | | | | |
| Health care worker in contact with patients | 30.0% | 36.1% | 27.0% | 0.211 |
| Clinical laboratory staff | 9.0% | 13.9% | 6.3% | |

Continuous and categorical variables are provided as median/interquartile ranges and percentages, respectively. P-values were derived using the two-sided Mann-Whitney U, Pearson χ2, or Fisher's exact tests to compare differences between groups as described in the Methods.

* History of exposure to COVID-19 was determined based on the combined S-IgG and S-IgA results (see the Methods and Results). Due to insufficient sample volume, two participants (2/100) were excluded from S-IgA testing; one of these participants was S-IgG+ and therefore included in the "Prior COVID-19" category despite their unknown S-IgA status.

~ Other, non-Kazakh, ethnic groups include people with Slavic and other Eastern European and Central Asian backgrounds.

#BMI data available for 97/100 participants.

^ Participants self-reported gastrointestinal conditions, hypertension, chronic heart disease, chronic obstructive pulmonary disease, history of malignancy, diabetes, liver disease, thyroid dysfunction, kidney disease, neurologic conditions, autoimmune conditions; the distribution of individual comorbidities did not differ between the "no prior COVID-19" and "prior COVID-19" groups.

for many other countries [7], albeit on par with the recent estimates from the neighbouring St. Petersburg, Russia, where antibodies against the receptor binding domain of the SARS-CoV-2 spike were detectable in ~45% of randomly sampled adults [14]. Similarly high SARS-CoV-2 seroprevalence rates have also been observed in the general populations of Brazil (>76% [15]), Ecuador (~45% [16]), India and Pakistan (>52% [17, 18]), and in communities at risk, such as healthcare workers and nursing home residents, across the globe [7]. Consistently, a recent analysis of serology data from a private laboratory network in Kazakhstan obtained for a period of 12 months (July 2020- July 2021) indicated a test positivity rate of 63% for SARS-CoV-2 IgG [19]. At the same time, a recently published household survey conducted across three cities in Kazakhstan between October 2020 and January 2021 found SARS-CoV-2 IgG/IgM-based seroprevalence rates to range from 39–61% [20].

In our cohort, serologically confirmed exposure to COVID-19 was associated with self-documented history of respiratory illness, most of which was dated by the participants to the peak of the first COVID-19 wave in the Spring-Summer of 2020 [1], period during which the country's healthcare system was overwhelmed and laboratory testing was limited [3]. This timing of self-reported illness suggests that anti-S immunoglobulins remain detectable up to a year after symptomatic COVID-19, consistent with the established long-term persistence of SARS-CoV-2-reactive antibodies [21, 22].

COVID-19 exposure in this cohort was significantly associated with Kazakh ethnicity, consistent with our earlier finding that Kazakh people are more likely have a laboratory-confirmed COVID-19 diagnosis but are less likely to develop severe disease compared to other ethnic groups in Kazakhstan [1]. Somewhat unexpectedly, we did not see any association between the serologically confirmed exposure to COVID-19 and the participants' professional occupation or any demographic factors. This may be because differences between sub-populations with different COVID-19 exposure risk are overwhelmed by the high seroprevalence in the general population. Alternatively, the risk of infection for healthcare workers may not be elevated relative to the general population because of the adequacy of infection control practices that are in place in healthcare facilities.

Given the logistical difficulties with procuring biomedical reagents and limited technological capacity in the setting of Kazakhstan, our choice of the serologic assay in this study was dictated by both assay quality and logistic feasibility. Therefore, we used a commercially available, FDA-approved assay, validated by several research groups, and deployable in a basic clinical lab setting [23–26]. Furthermore, we chose to use both IgG and IgA based on the evidence of distinct but overlapping temporal patterns seen for these antibodies in COVID-19 patients [21, 27, 28]. Thus, both IgG and IgA appear as early as 2 weeks post-symptom onset (PSO), with IgA increasing up to third week PSO and then dropping, while IgG increases until fourth week PSO, remaining detectable up to 8 months PSO [21]. Finally, we chose not to use IgM, since this antibody is more suitable for detecting acute infection, while in convalescent subjects IgA temporally overlaps IgM, resulting in higher positivity rates [27].

We validated our findings in the current cohort by testing pre-pandemic samples and performing virus neutralization assays. Half of the Prior COVID participants exhibited virus neutralization correlating with S-reactive IgG titres. This finding is consistent with the evidence that SARS-CoV-2 neutralization declines rapidly after COVID-19 disease resolution and >40% of convalescent subjects show little neutralization activity [21, 23, 29].

Recent studies indicate that people with prior history of COVID-19 have a stronger response to vaccination compared to COVID-19-naive subjects after one vaccine dose [30–32]. Mass COVID-19 vaccination was launched in Kazakhstan in February 2021, and so far, ~40% of Kazakhstan's population has received at least one vaccine dose [10]. Considering limited vaccine supply and low vaccination acceptance, our seroprevalence findings therefore

could be extended to inform the ongoing vaccination efforts about the existing population-wide anti-SARS-CoV-2 immunity in Kazakhstan. For example, the second dose of COVID-19 immunisation could be reserved for people without prior natural exposure to SARS-CoV-2 but delayed for subjects with prior COVID-19 exposure.

Given the small sample constrained to employees of one, albeit large (employing ~3000 staff) organization, our findings should be seen as preliminary "pilot" data on SARS-CoV-2 prevalence in Kazakhstan. Importantly, our analysis was focused on S-reactive immunoglobulins and with increasing vaccination coverage other SARS-CoV-2 antigens should be incorporated into the future seroprevalence surveys across various demographic groups in the region.

## Conclusions

Continuous epidemiologic surveillance of SARS-CoV-2 exposure is critical for understanding the COVID-19 transmission dynamics and for informing ongoing vaccination efforts and other COVID-19 mitigation strategies. Seroprevalence studies could help fill in the current gaps in COVID-19 case reporting in Kazakhstan. However, large scale seroprevalence studies in resource-limited settings may not be feasible due to limited clinical and laboratory infrastructure- a barrier that could be overcome by using alternative approaches such as analysing blood bank samples, assessing data collected by the private laboratory sector (which has recently dramatically expanded in Kazakhstan [19]), and implementing PCR-based wastewater surveillance. Our seroprevalence study for the first time documents an extremely high rate of SARS-CoV-2 exposure in Karaganda, Kazakhstan. Although pilot in nature, these findings are consistent with high SARS-CoV-2 seroprevalence in other regions of Kazakhstan [19, 20, 33], corroborating an overall underascertainment of COVID-19 rates across the country. While narrowing the gaps in country-wide infection surveillance necessitates addressing systemic issues related to governance and healthcare delivery and improving vital registration systems, we hope that our findings can inform the regional pandemic response to facilitate a more effective and equitable distribution of resources, such as vaccines.

## Supporting information

**S1 Dataset. Raw study dataset.**
(XLSX)

## Acknowledgments

We thank all the study participants and the COVID-19 vaccination clinic staff. We acknowledge that an earlier draft of this manuscript appeared online as a medRxiv preprint [5].

## Author Contributions

**Conceptualization:** Irina Kadyrova, Sergey Yegorov, Dmitriy Babenko.

**Data curation:** Irina Kadyrova, Sergey Yegorov, Ilya Korshukov, Dmitriy Babenko.

**Formal analysis:** Irina Kadyrova, Sergey Yegorov, Ilya Korshukov, Dmitriy Babenko.

**Funding acquisition:** Irina Kadyrova, Dmitriy Babenko.

**Investigation:** Irina Kadyrova, Sergey Yegorov, Baurzhan Negmetzhanov, Yevgeniya Kolesnikova, Svetlana Kolesnichenko, Ilya Korshukov, Dmitriy Vazenmiller, Yelena Stupina, Naylya Kabildina, Assem Ashimova, Aigul Raimbekova, Anar Turmukhambetova, Matthew S. Miller, Gonzalo Hortelano, Dmitriy Babenko.

**Methodology:** Irina Kadyrova, Sergey Yegorov, Baurzhan Negmetzhanov, Yevgeniya Kolesnikova, Svetlana Kolesnichenko, Ilya Korshukov, Lyudmila Akhmaltdinova, Dmitriy Vazenmiller, Yelena Stupina, Naylya Kabildina, Matthew S. Miller, Dmitriy Babenko.

**Project administration:** Irina Kadyrova, Matthew S. Miller, Gonzalo Hortelano, Dmitriy Babenko.

**Resources:** Irina Kadyrova, Anar Turmukhambetova, Gonzalo Hortelano.

**Supervision:** Irina Kadyrova, Gonzalo Hortelano.

**Validation:** Sergey Yegorov.

**Visualization:** Sergey Yegorov.

**Writing – original draft:** Irina Kadyrova, Sergey Yegorov.

**Writing – review & editing:** Irina Kadyrova, Sergey Yegorov, Yevgeniya Kolesnikova, Svetlana Kolesnichenko, Ilya Korshukov, Lyudmila Akhmaltdinova, Assem Ashimova, Aigul Raimbekova, Anar Turmukhambetova, Matthew S. Miller, Gonzalo Hortelano.

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
