## [Decision Letter · Decision Letter 0]

17 Jun 2022

PONE-D-22-14174High seroprevalence of SARS-CoV-2 antibodies in Karaganda, Kazakhstan before the launch of COVID-19 vaccination.PLOS ONE

Dear Dr. Yegorov,

Thank you for submitting your manuscript to PLOS ONE. After careful consideration, we feel that it has merit but does not fully meet PLOS ONE’s publication criteria as it currently stands. Therefore, we invite you to submit a revised version of the manuscript that addresses the points raised during the review process.

Carefully consider the comments of the reviewers, which provide important points for improvement, and also the following:

- Briefly explain the purpose of the pre-pandemic group in the Methods as well;

- Explain which tests were used for which variables (instead of 'as appropriate');

- Please kindly provide the reference number of the ethical clearance for the larger study within which this serologic survey was performed;

- Results are given for 99 subjects in the Table and for 98 subjects in the Results, please either correct or explain this discrepancy;

- It is not clear whether 55 (methods) or 35 (results) subjects were selected for the neutralization assay, and by which criterion;

- Please only use percentages in the table, as the numbers can be inferred thanks to the total in the top row (you can write "Results are expressed as percentages unless specified otherwise" in the title of the table; consistently, specify that for continuous variables you are providing the mean (and SD?); finally, the tests used should be explained as a note, not within the title of the table.

We look forward to receiving your revised manuscript.

Kind regards,

Cecilia Acuti Martellucci, M.D.

Academic Editor

PLOS ONE

Journal Requirements:

"The study was funded by the Ministry of Education and Science of the Republic of Kazakhstan (AP09259123) and, in part, by the Nazarbayev University grant #280720FD1902 to GH. SY was supported, in part, by a M.G. DeGroote Postdoctoral Fellowship."

"The study was funded by the The study was funded by the Ministry of Education and Science of the Republic of Kazakhstan (AP09259123, https://www.gov.kz/memleket/entities/edu?lang=en) and, in part, by the Nazarbayev University (https://nu.edu.kz/) grant #280720FD1902 to GH. SY was supported, in part, by a M.G. DeGroote Postdoctoral Fellowship.  (AP09259123) and, in part, by the Nazarbayev University grant #280720FD1902 to GH. SY was supported, in part, by a M.G. DeGroote Postdoctoral Fellowship. The funder of the study had no role in study design, data collection, data analysis, data interpretation, or writing of the report. All authors had full access to all the data in the study and the lead authors (IK, SY, DB) had final responsibility for the decision to submit manuscript for publication."

Reviewers' comments:

Reviewer's Responses to Questions

**Comments to the Author**

1. Is the manuscript technically sound, and do the data support the conclusions?

Reviewer #1: Yes

Reviewer #2: Yes

2. Has the statistical analysis been performed appropriately and rigorously? 

Reviewer #1: Yes

Reviewer #2: Yes

3. Have the authors made all data underlying the findings in their manuscript fully available?

Reviewer #1: Yes

Reviewer #2: Yes

4. Is the manuscript presented in an intelligible fashion and written in standard English?

Reviewer #1: Yes

Reviewer #2: Yes

5. Review Comments to the Author

Reviewer #1: The paper is well written and is relevant to current situation. Coming from one of the “stans” myself I realize the level of underreporting for such data. This makes it easier for me to understand the situation and importance of this paper. However, I have several concerns:

1. The history and situation with under reporting in the region is not fully explained.

2. The cohort chosen cannot be generalized to the whole population - this is mentioned in the paper, but I think it would be good to provide more details about reasons for choosing this particular cohort (availability, connections, etc)

3. Last, but not least, I felt that there is something missing in conclusion. Specifically, lack of recommendation for further studies and possible actions. It was established that the prevalence of COVID-19 in the region is much higher than officially reported. What can be recommended to the government, how to improve reporting in general, how this underreporting is connected to uptake of vaccinations?

If possible this information can be added to the paper to make it more convenient complete.

Reviewer #2: Dear Authors!

On acquainting with the presented paper, I found no significant remarks in the Methodology, Results presentation, and the Discussion section.

The topic's relevance cannot be considered overestimated, although the tension of the Covid-19 pandemic has significantly decreased. Given the chance of the possible upcoming pandemics, the global scientific community strives to acquire a deeper understanding of all the details of the present pandemic process. In this relation, information from the countries with a comparatively low level of healthcare, such as Kazakhstan (LMICs out of the EU), appears to be particularly valuable.

A study on Sputnik V preceded this work, and linked preprints from the MedRxiv Yale were placed into the References.

In the methodology narration, the solution to use the manufacturers' cutoff values as being more suitable was relevant, thus indicating the quality of the work done.

The dubious moment was the number of pre-pandemic samples (N 10), though it seems that this circumstance did not interfere with the validity of the results obtained.

The only I would suggest is to change your title slightly by inserting the brackets to lessen the number of commas:

"High seroprevalence of SARS-CoV-2 antibodies in Karaganda (Kazakhstan) before the launch of COVID-19 vaccination"

I found this conclusion the most important: "...serologically assessed SARS-CoV-2 exposure in this cohort was 14-15-fold higher than the reported all-time national and regional COVID-19 prevalence; the unusually high SARS-CoV-2 seroprevalence exceeding the estimates for many other countries."

These data allow understanding of the shortcomings and pitfalls in the preventive strategies against the pandemic held in Kazakhstan. For instance, the paper showed the extremely low effectiveness of the screening by currently applied means. Albeit all the screened samples were negative, 59.2% of them turned out to be positive for anti-S IgA.

I highly appreciated the integrity in presenting your data.

6. PLOS authors have the option to publish the peer review history of their article (what does this mean?). If published, this will include your full peer review and any attached files.

Reviewer #1: No

Reviewer #2: No

---

## [Author Response · Author response to Decision Letter 0]

28 Jun 2022

Reviewer #1: The paper is well written and is relevant to current situation. Coming from one of the “stans” myself I realize the level of underreporting for such data. This makes it easier for me to understand the situation and importance of this paper. However, I have several concerns:

Authors' response: We are very appreciative of the reviewer’s positive view of our work and thank the reviewer for the insightful comments and questions. Please kindly find our point-by-point responses to the reviewer’s remarks below.

1. The history and situation with under reporting in the region is not fully explained.

Authors' response: Thank you for raising this point. To address this comment, we have now added additional information to the Introduction to describe the situation with regard to COVID-19 under-reporting in Kazakhstan (p3, lines 15-23 and p4, lines 1-3).

2. The cohort chosen cannot be generalized to the whole population - this is mentioned in the paper, but I think it would be good to provide more details about reasons for choosing this particular cohort (availability, connections, etc)

Authors' response: Thank you for this very valid comment. This cohort was chosen for the serologic studies owing to funding availability and perceived feasibility in the context of the readily available resources and active participant recruitment within the infrastructure of the larger clinical trial. We have now added this information to the Materials and Methods (p4, lines 15-18).

3. Last, but not least, I felt that there is something missing in conclusion. Specifically, lack of recommendation for further studies and possible actions. It was established that the prevalence of COVID-19 in the region is much higher than officially reported. What can be recommended to the government, how to improve reporting in general, how this underreporting is connected to uptake of vaccinations? If possible this information can be added to the paper to make it more convenient complete. 

Authors' response: Thank you for this suggestion! We have modififed the Discussion/Conclusions (p 12, lines 9-23) to include our recommendations with regard to improving case reporting and pandemic surveillance using serological screening and alternative approaches.

Reviewer #2: Dear Authors! On acquainting with the presented paper, I found no significant remarks in the Methodology, Results presentation, and the Discussion section. The topic's relevance cannot be considered overestimated, although the tension of the Covid-19 pandemic has significantly decreased. Given the chance of the possible upcoming pandemics, the global scientific community strives to acquire a deeper understanding of all the details of the present pandemic process. In this relation, information from the countries with a comparatively low level of healthcare, such as Kazakhstan (LMICs out of the EU), appears to be particularly valuable. A study on Sputnik V preceded this work, and linked preprints from the MedRxiv Yale were placed into the References. In the methodology narration, the solution to use the manufacturers' cutoff values as being more suitable was relevant, thus indicating the quality of the work done. The dubious moment was the number of pre-pandemic samples (N 10), though it seems that this circumstance did not interfere with the validity of the results obtained. 

The only I would suggest is to change your title slightly by inserting the brackets to lessen the number of commas: "High seroprevalence of SARS-CoV-2 antibodies in Karaganda (Kazakhstan) before the launch of COVID-19 vaccination" I found this conclusion the most important: "...serologically assessed SARS-CoV-2 exposure in this cohort was 14-15-fold higher than the reported all-time national and regional COVID-19 prevalence; the unusually high SARS-CoV-2 seroprevalence exceeding the estimates for many other countries." These data allow understanding of the shortcomings and pitfalls in the preventive strategies against the pandemic held in Kazakhstan. For instance, the paper showed the extremely low effectiveness of the screening by currently applied means. Albeit all the screened samples were negative, 59.2% of them turned out to be positive for anti-S IgA. I highly appreciated the integrity in presenting your data. 

Authors' response: We thank the reviewer for a very thoughtful and positive review of our work! 

We agree with the reviewer that the original title of the manuscript appeared somewhat cumbersome, therefore we have adjusted it to: “High SARS-CoV-2 seroprevalence in Karaganda, Kazakhstan before the launch of COVID-19 vaccination.” We would be happy to replace the comma with brackets and ultimately leave this decision to the reviewers' discretion. 

Editorial Comments:

- Briefly explain the purpose of the pre-pandemic group in the Methods as well;

Authors' response: Done, please see p 5, lines 8-11.

- Explain which tests were used for which variables (instead of 'as appropriate');

Authors' response: Done, please see p 7, lines 13-18.

- Please kindly provide the reference number of the ethical clearance for the larger study within which this serologic survey was performed;

Authors' response: Done, we have added this information to the "Ethics Statement" (p 7, lines 19-22).

- Results are given for 99 subjects in the Table and for 98 subjects in the Results, please either correct or explain this discrepancy;

Authors' response: We apologize for the the lack of clarity here. Two out of 100 participants were excluded from S-IgA testing due to insufficient sample volume. However, because one of these two participants tested S-IgG+, they were still included in the "Prior COVID-19" category" despite the missing S-IgA result. We have now made edits to both the Table (see Table 1 footnote) and the Results (p.8, lines 8-16) to clarify this aspect.

- It is not clear whether 55 (methods) or 35 (results) subjects were selected for the neutralization assay, and by which criterion;

Authors' response: We are sorry for the confusion here. The neutralization assays were performed on 55 samples (of which 35 were classified as Prior COVID based on the S-IgG and S-IgA ELISA), as stated in the Methods. We have now clarified this in the Results (p8, lines 7-19). 

- Please only use percentages in the table, as the numbers can be inferred thanks to the total in the top row (you can write "Results are expressed as percentages unless specified otherwise" in the title of the table; consistently, specify that for continuous variables you are providing the mean (and SD?); finally, the tests used should be explained as a note, not within the title of the table.

Authors' response: Thank you for these suggestions. We have edited the Table as requested by the Editor and among other edits specified in the Table footnote the following: "Continuous and categorical variables are provided as median/interquartile ranges and percentages, respectively."

---

## [Decision Letter · Decision Letter 1]

12 Jul 2022

High SARS-CoV-2 seroprevalence in Karaganda, Kazakhstan before the launch of COVID-19 vaccination.

PONE-D-22-14174R1

Dear Dr. Yegorov,

We’re pleased to inform you that your manuscript has been judged scientifically suitable for publication and will be formally accepted for publication once it meets all outstanding technical requirements.

Kind regards,

Cecilia Acuti Martellucci, M.D.

Academic Editor

PLOS ONE

Additional Editor Comments (optional):

Reviewers' comments:

Reviewer's Responses to Questions

**Comments to the Author**

1. If the authors have adequately addressed your comments raised in a previous round of review and you feel that this manuscript is now acceptable for publication, you may indicate that here to bypass the “Comments to the Author” section, enter your conflict of interest statement in the “Confidential to Editor” section, and submit your "Accept" recommendation.

Reviewer #1: All comments have been addressed

Reviewer #2: All comments have been addressed

2. Is the manuscript technically sound, and do the data support the conclusions?

Reviewer #1: Yes

Reviewer #2: Yes

3. Has the statistical analysis been performed appropriately and rigorously? 

Reviewer #1: Yes

Reviewer #2: Yes

4. Have the authors made all data underlying the findings in their manuscript fully available?

Reviewer #1: Yes

Reviewer #2: Yes

5. Is the manuscript presented in an intelligible fashion and written in standard English?

Reviewer #1: Yes

Reviewer #2: Yes

6. Review Comments to the Author

Reviewer #1: (No Response)

Reviewer #2: That's quite an improvement, and I am satisfied with all corrections made. I also agree with your present title.

7. PLOS authors have the option to publish the peer review history of their article (what does this mean?). If published, this will include your full peer review and any attached files.

Reviewer #1: No

Reviewer #2: No

---

## [Editor Report · Acceptance letter]

18 Jul 2022

PONE-D-22-14174R1 

High SARS-CoV-2 seroprevalence in Karaganda, Kazakhstan before the launch of COVID-19 vaccination. 

Dear Dr. Yegorov:

I'm pleased to inform you that your manuscript has been deemed suitable for publication in PLOS ONE. Congratulations! Your manuscript is now with our production department. 

Kind regards, 

on behalf of

Dr. Cecilia Acuti Martellucci 

Academic Editor

PLOS ONE